# Modelling of Electromagnetic Energy Harvester with Rotational Pendulum Using Mechanical Vibrations to Scavenge Electrical Energy

**Bartłomiej Ambrożkiewicz \*, Grzegorz Litak and Piotr Wolszczak** 

Department of Automation, Faculty of Mechanical Engineering, Lublin University of Technology, Nadbystrzycka 36, 20-618 Lublin, Poland; g.litak@pollub.pl (G.L.); p.wolszczak@pollub.pl (P.W.)
\* Correspondence: b.ambrozkiewicz@pollub.pl

**Featured Application: A concept of a non-linear electromagnetic energy harvesting (EH) system with a rotational pendulum was proposed. In the system, non-linearity was introduced through the interaction of additional stationary magnets placed into coils with a tip magnet on the pendulum by creating double potential well. In the system, both oscillations and rotations are possible depending on the system parameters including the strength of magnets, the excitation conditions including amplitude and frequency, and the initial conditions. Moreover, a housing for an EH prototype was made in a 3D printing technique in order to avoid undesirable electromagnetic coupling. Data obtained from a finite element method (FEM) model, real data, and data from an experiment were applied to reflect the system's operation. The proposed EH device can be used in applications, where a kinematic longitude excitation with fairly small frequency is available.**

**Abstract:** A concept of non-linear electromagnetic system with the rotational magnetic pendulum for energy harvesting from mechanical vibrations was presented. The system was stimulated by vertical excitation coming from a shaker. The main assumption of the system was the montage of additional regulated stationary magnets inside coils creating double potential well, and the system was made with a 3D printing technique in order to avoid a magnetic coupling with the housing. In validation process of the system, modelling of electromagnetic effects in different configurations of magnets positions was performed with the application of a finite element method (FEM) obtaining the value of magnetic force acting on the pendulum. A laboratory measurement circuit was built and an experiment was carried out. The voltage and power outputs were measured for different excitations in range of system operational frequencies found experimentally. The experimental results of the physical system with electrical circuit and numerical estimations of the magnetic field of a stationary magnet's configuration were used to derive a mathematical model. The equation of motion for the rotational pendulum was used to prove the broadband frequency effect.

**Keywords:** energy harvesting; rotational pendulum; electromagnetism; mathematical model

---

## 1. Introduction

Through the last decade, research on a new field of activity, energy harvesting (EH), has gained recognition both in scientific and industrial environments. It denotes the effort to develop small distributed renewable sources of energy and the need to power up low-voltage autonomous electrical devices and batteries. Many EH devices have already found application in a few technical solutions i.e., automotive [1], wireless sensor networks (WSN) [2], health-monitoring devices [3], and civil

engineering [4]. The main advantage of the aforementioned device is its substitutability for a battery as a source of energy. The energy is scavenged from the environment, examples of which can be solar energy, tides, gravitational force or mechanical vibrations [5]. Most of the currently designed systems are electromagnetic, piezoelectric and electrostatic [5–11].

From the application point of view, research aimed at increasing the density of recovered energy is important for systems generating small portions of energy in a fairly wide frequency band [12–14]. That is why energy harvesting systems with non-linear behavior are researched [15–20]. The main advantages of non-linearities introduced to the system are the possibilities to modify the resonance behavior [21,22], to obtain good off-resonance performance [23–25] or to attain multi-resonance [26].

Our proposed concept is based on an electromagnetic system with a rotational pendulum. Energy harvesters with a rotational pendulum as a resonator can evolve in time with a large amplitude. Earlier works on the system's development started from theoretical analysis of its dynamics and studies on its response by different excitation conditions [27–32]. Next, several systems were built, optimized and tested for different applications, primarily energy harvesting from human motion [33–37] and automotives [38,39]. In our system we focused on introducing double potential well by mounting adjustable additional magnets inside coils expecting non-linear behavior brought by magnetic interactions. The magnet's position and its repelling alignment will enable pendulum's rotation only by specific excitation frequency which was found experimentally. To obtain a pure system's response, the corresponding parts were made with a 3D printing technique. Some improvements connected with design had to be introduced after performing tests. The system was validated with mathematical model and variables used in the pendulum's equation of motion were based on simulation and experiment. Voltage and power outputs were measured with various excitation frequencies.

## 2. Materials and Methods

For the implementation of a specific research task, the idea of electromagnetic system with a rotational pendulum for energy harvesting from mechanical vibrations has been proposed. A prototype of the device was made for carrying out a physical experiment and a model for carrying out numerical simulations was developed. The experiment's methodology started with the performance of the finite element method (FEM) model, where the value of the magnetic force acting on a pendulum was found. The physical experiment provided information on operational frequency range, acceleration amplitude acting on the system during test and rotational velocity of a pendulum. Values found in the experiment were used in formulation equation of motion. Other values were calculated from system's geometry, such as moment of inertia and distance from pivot to the center of gravity. The system's operation was validated with the adjusted mathematical model.

### 2.1. Prototype and Physical Experiment

The prototype of the energy harvester with a rotational pendulum is based on electromagnetic induction phenomena. To rid it of magnetic coupling, the system's housing and pendulum's housing are made of ABS (Acrylonitrile Butadiene Styrene) in 3D printing technique (Figure 1). Many steps are taken to miniaturize the EH systems, because in case of electromagnetic systems this is hard to do. Our goal was not to exceed 10 cm of system's design in any direction, what was succeeded. The system consists of 12 coils connected in series mounted symmetrically on both sides (6 on one side) of the energy harvester. By such a connection we wanted to obtain the biggest possible voltage/power response from the system. The system's design provide to apply versatile coils connection, exemplary connecting 6 coils mounted on one wall of the system. Inside 4 of them, additional adjustable permanent magnets are installed, two of them on each side of the system (Figure 2). They will have an influence on electromagnetic induction of the system and movement of the pendulum by creating double potential well. Because the initial design of the pendulum was too light, additional inertial masses had to be mounted on the system's sides to increase its moment of inertia. The total weight of entire system was equal to ca. 900 g. In the system, both mechanical and electrical damping will occur. Mechanical

damping was in form of friction inside bearings, and an electrical one from additional resistors. The first one was only considered later in the formulation of the equation of motion.

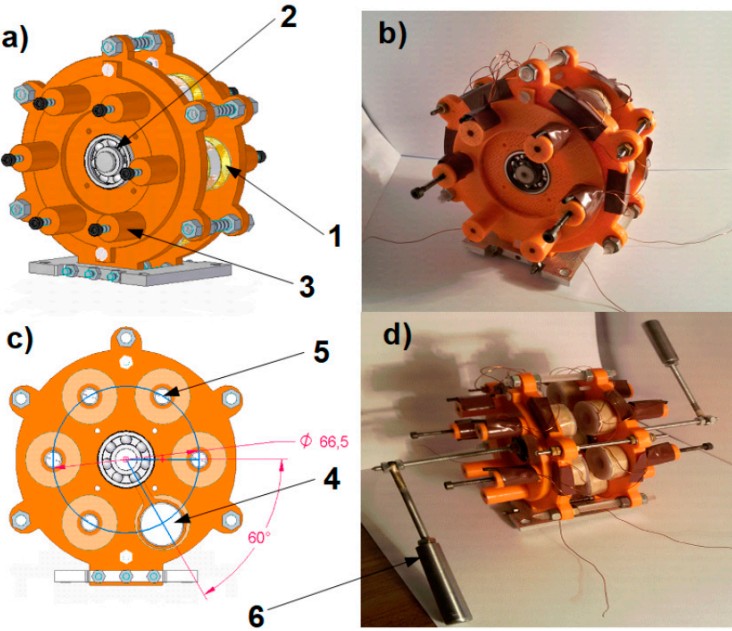

**Figure 1.** (**a**) Solid model and real system of the electromagnetic energy harvester: (1) coil, (2) ball bearing, (3) longitudinal, cylindrical free space in form channels for additional magnets, (**b**) system made using 3D printing technique, (**c**) cross-section of system: (4) main pendulum with magnet, (5) exemplary additional magnet mounted inside coil, (**d**) system with additional inertial masses: (6) additional external inertial masses of the pendulum.

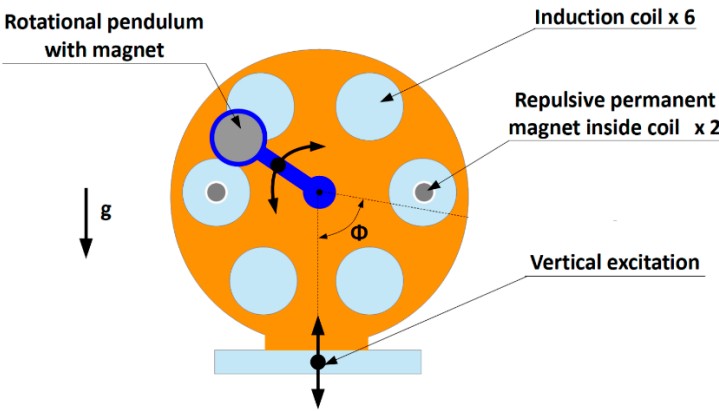

**Figure 2.** Principle of operation and alignment of coils and additional magnets. Interior view of the system in a plane perpendicular to the axis of rotation of the pendulum. The system could be fixed to the moving frame with an additional constant tilting angle.

In the prototype presented above it is possible to adjust a few factors such as the position of additional magnets, additional inertial masses of pendulum and the resistance load attached to the electric circuit. The aforementioned variables will have an influence on output results: voltage, power, magnetic force and magnetic induction in the system. Change of magnetic force and electromagnetic induction were simulated in Finite Element Method Magnetics (FEMM) software. The principle of calculations will be explained in next sections. Because we used the strongest available magnets made of neodymium N37 for the magnetic forces acting between them, in the experiment, results were only obtained for the furthest distance between them. Then, the additional load was regulated and output

voltage and power were measured. Such measurements were performed for different frequencies of excitation to obtain chart output voltage/power in the frequency of excitation domain.

The scheme of measuring setup is presented in Figure 3. The measurement system consisted of the following equipment: signal generator—KZ 1406 (ZOPAN, Warsaw, Poland), signal amplifier—BAA 120 (TIRA GmbH, Schalkau, Germany), vibration exciter—S 513 (TIRA, Schalkau, Germany), oscilloscope—MDO3024 (Tektronix, Beaverton, OR, USA), accelerometer—AltIMU-10 v3 (Pololu, Las Vegas, NV, USA). Entire measuring system starts with signal generator, where periodic sinusoidal signal was generated with specific frequency matched to the operational frequency of the energy harvester, where the rotational movement of the pendulum could be sustained. Next, the signal was processed with a signal amplifier, and such a signal was steering the shaker's movement exciting the energy harvester. Signals of output voltage and output power were transferred to the oscilloscope. Different values of output voltage and power were obtained by additional resistance in the form of a potentiometer. On the energy harvester housing, an accelerometer was mounted, the measured amplitude of acceleration acting on the system was equal to 0.55 g, by input current equal to 1.1 A, and input voltage of 4.0 V displayed on the amplifier. The value of acceleration amplitude was taken into the mathematical model. Below in Tables 1 and 2, the most important characteristics of used vibration exciter and accelerometer are given.

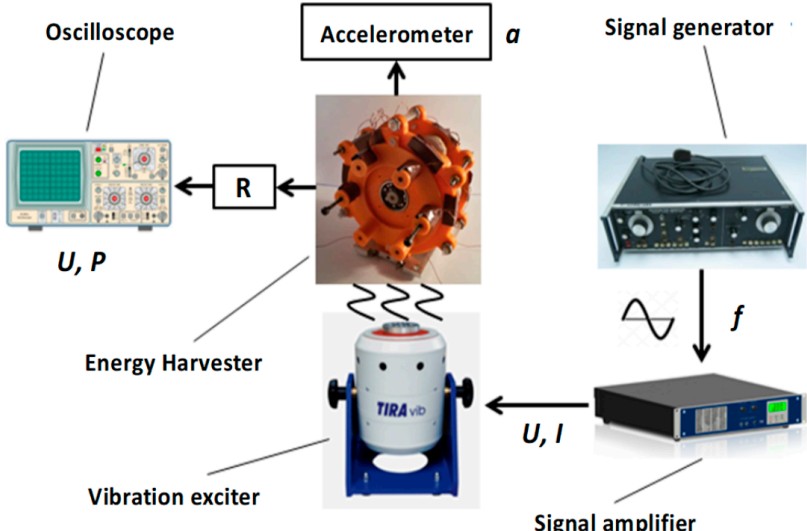

**Figure 3.** Scheme of measuring setup.

**Table 1.** Characteristics of the vibration exciter TIRA S 513.

| Feature | Value |
|---|---|
| Rated peak force [N] | 100 |
| Frequency range [Hz] | 2–7000 |
| Max. rated travel [mm] Pk-Pk | 13 |
| Max. velocity [m/s] sine | 1.5 |
| Max. acceleration [g] sine | 45 |

**Table 2.** Characteristics of the accelerometer AltIMU-10 v3.

| Feature | Value |
|---|---|
| Operating voltage [V] | 2.5–5.0 |
| Supply current [mA] | 6 |
| Output format | I$^2$C |
| Sensitivity range [g] | ±2, ±4, ±6, ±8, ±10 |

## 2.2. Numerical Simulations

The numerical calculations of electromagnetic induction and magnetic force values were performed using the finite element method. Namely, the FEMM software (Ph.D. David Meeker, Waltham, MA, USA) was used, which consider magnetic, electrostatic, heat flow and current flow problems. The first simulation task was the change of magnetic induction value in the angular and linear position change of the additional magnet inside the coils and mounted on the pendulum. Then, we received results in the form of a 3D surface presenting the change of electromagnetic phenomena in the system. The estimation was performed for a coil, an additional permanent magnet and the permanent tip magnet mounted on the rotating pendulum keeping the geometrical and magnetic features of applied magnets (NdFeB 37) and coils (28 AWG); also, the repulsive orientation of magnets was taken into account. The aim of the simulation is to check how the change of additional magnet position influence on magnetic induction arose in the coil [40–42]. In the Figure 4 the layout of the magnets and coil and exemplary results in FEMM software are presented.

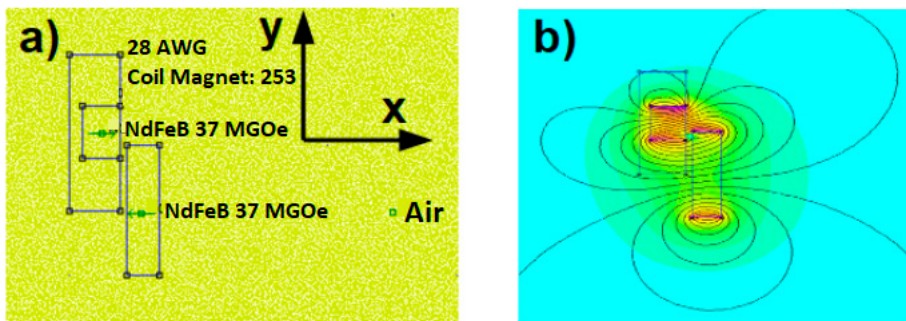

**Figure 4.** (**a**) Magnets and coil layout in Finite Element Method Magnetics (FEMM) pre-processor, (**b**) Exemplary results in FEMM solver.

In total, created surface will consist of 900 results, changing the position of magnet inside coil by 0.5 mm and position of magnet on pendulum by 2.5°. The resultant value of electromagnetic induction is taken from the coil in the *x*-axis. The magnetic induction results surface is presented in Figures 5 and 6.

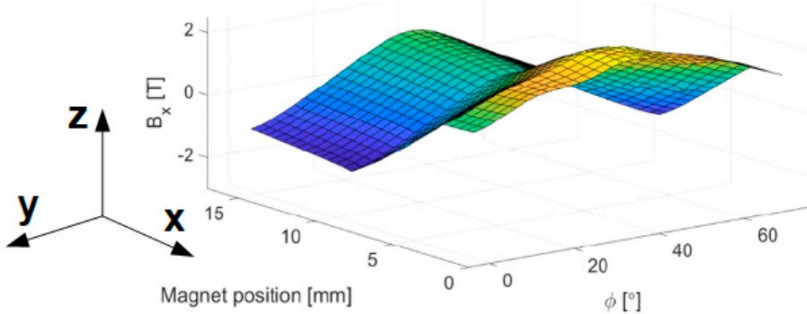

**Figure 5.** Magnetic induction results in *x*-axis in additional magnet position and angular magnet position domain.

The position of the additional magnet was limited by the geometry of the plastic cylindrical channels on the sides of the system. Value discrepancies of magnetic induction are the result of magnetic interactions between magnets. It must be emphasized that only two pairs of parallel coils with additional magnets were installed. In other coils without additional magnets, the value of electromagnetic induction would be smaller.

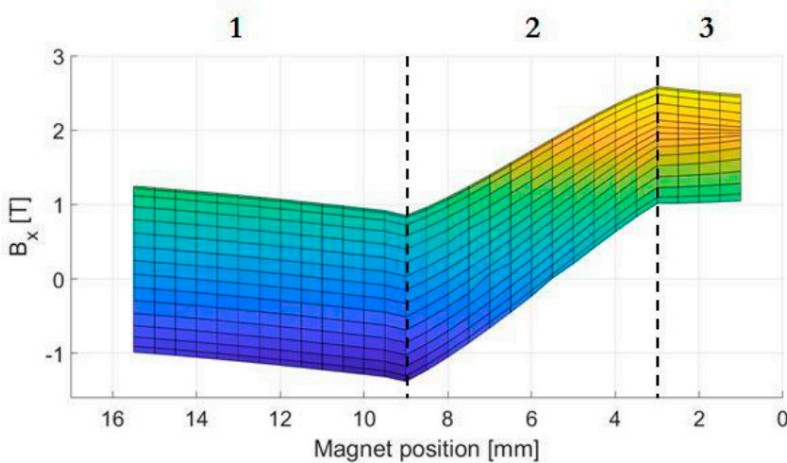

**Figure 6.** Magnetic induction in *x*-axis in additional magnet position domain. Three specific areas of interests were found: 1—additional magnet fully outside coil, 2—additional magnet partially inside in and outside coil, 3—additional magnet fully inside coil.

The second simulation task using the FEMM software was to find the value of magnetic force between acting magnets and use such value for modelling of mechanical system. The manual of FEMM software predicts the application of co-energy of entire energetic system for two different close positions [43]. The magnetic force equation has the following form:

$$F_{mag} = \frac{-\delta W_c}{\delta l},$$

(1)

where $\delta W_c$ is variation of co-energy value of entire energetic system for two different close positions, $F_{mag}$ is the magnetic force acting on the pendulum, $\delta l$ is the variation of the additional magnet position.

To compare the magnetic forces' values in the system for different distances between additional magnet and the pendulum, Figure 7 shows the curves for x = 1 mm and x = 13 mm. The differences between the curves in Figure 7 are caused by the magnetic interactions. The shape with three local maxima results from the rectangular shape of cross-section of the magnets in 2D simulation. The amplitude and distribution of local maxima changes with the distance between the magnets. In Figure 7b, the local maxima disappear due to the decrease in amplitude of the magnetic force for the 13 mm distance. Two-dimensional simulation provided very important information concerning change of magnetic force depending on the distance between magnets and the obtained values can be used in the equation of motion.

The calculated resultant value of magnetic force acting on the pendulum was applied to the equation of motion of the system. The aforementioned magnetic force comprised in some way the damping of the pendulum's movement. That is why its value is so important in modelling. Because of the appearance of magnetic forces that were too strong by close relative alignment of the magnets, in the experiment and in the mathematical model only the long distance between the magnets was considered.

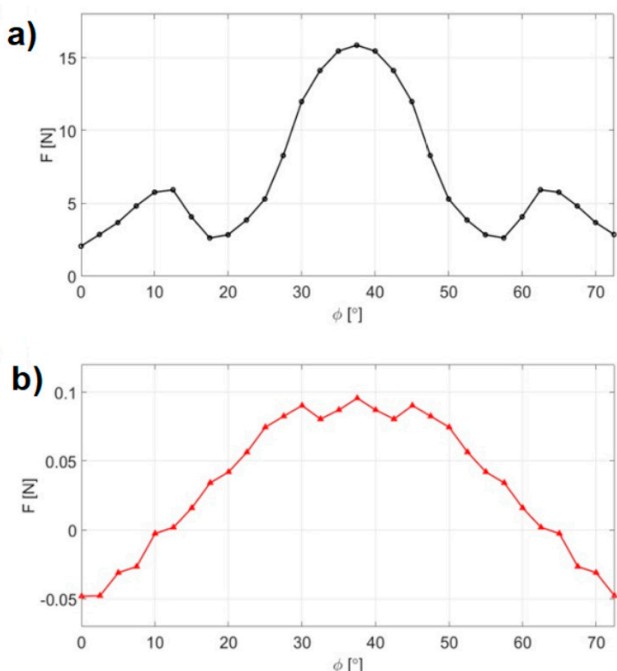

**Figure 7.** Comparison of total magnetic forces values in the system for: (**a**) x = 1 mm (**b**) x = 13 mm, x is the distance between the additional magnet and the pendulum. Note that, the secondary peaks visible in Figure 7a originated from the finite size of magnet dipoles.

## 3. Experimental Results

Using the measuring system described in the previous section, measurements of output voltage and output power were performed in the domain of variable operating frequency [44,45]. Also measurements were conducted in open loop (directly, without additional resistance). First tests showed that the movement of the pendulum is supported by frequencies from 10 Hz to 15 Hz. Experimental tests gave important information about which environment the system can succeed in. The disadvantage can be short broadband of operational frequency, but it is possible to broaden it by application of heavier masses on additional inertial pendulums. Output signals were displayed on the oscilloscope and its results were stored. Then, visualization and proper calculations were performed in Matlab software. Measured values of output voltage were stored with a sampling rate of 100,000 *S/s* per channel. For the tests, seven different values of additional resistance were considered mainly close to the value of internal resistance of 12 coils in the system connected in series which value was 21.5 Ω. The most relevant characteristics of performed tests are collected in Table 3.

**Table 3.** Characteristics of the test.

| Test Feature | Input Data |
|---|---|
| Operational frequencies [Hz] | 10; 11; 12; 13; 14; 15 |
| Internal resistance of the system [Ω] | 21.5 |
| Additional resistance [Ω] | 3.3; 6.5; 12.3; 17; 21.5; 25.7; 30.3 |
| Sampling rate per channel [*S/s*] | 100,000 |

After collecting the raw signal of output voltage, we calculated its root mean square value (2). After such calculations output power was calculated (3). All the results were visualized in the operating frequency domain (Figure 8).

$$U_{RMS} = \sqrt{\frac{1}{N}\sum_{k=0}^{N-1} u(k)^2} \tag{2}$$

$$P = \frac{U_{RMS}^2}{R} \tag{3}$$

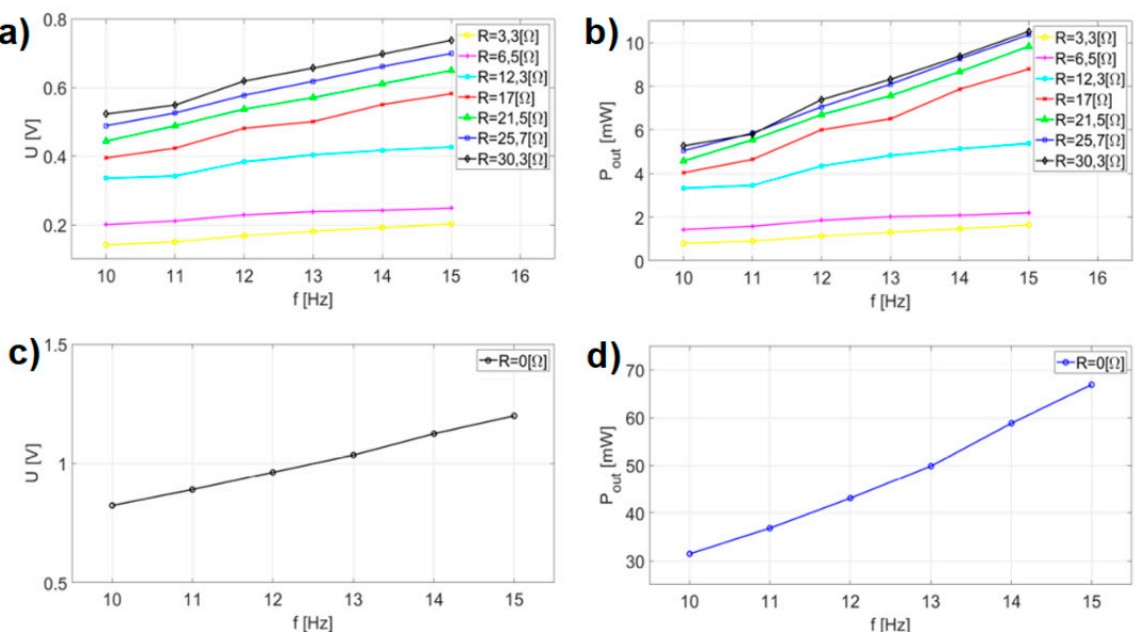

**Figure 8.** Results of (**a**) root mean square (RMS) voltage in the operating frequency domain for different values of additional resistance, (**b**) output power in the operating frequency domain for different values of additional resistance, (**c**) RMS voltage in the operating frequency domain for open loop, (**d**) output power in operating frequency domain for open loop.

An increase of the additional resistance value causes the increase of the voltage RMS value from the excited system. However, it can be expected that the mentioned value would not be as great and reach almost the limit of 30.3 Ω. Cross-checking measurements must be performed for higher values of additional resistance or by wider range of operating frequency. Unfortunately, design constraints hindered it. The results of RMS voltage value in open loop show its linear change with operational frequency. Tests with the open loop are not in practical usage, but they can show the limits of the system. The above presented results were only performed for the frequency range, where rotational movement was supported. To expand it, pendulum's moment of inertia must be changed. Moreover, for the optimization of the system's power efficiency, extended experiments can be performed in wider range of additional resistance [46].

## 4. Mechanical Model of the Mechanical Resonator and a Broadband Effect

Research on small portions of mechanical energy conversion into electrical energy was started from mechanical vibrations. Results obtained were adjusted to needs of specific energy receiver paying special attention for output power, resonance frequency or energy conversion. Continuous development favors usage of new materials and systems managing conversion and flow of energy to the receiver. Nowadays, in the beginning of work on the design of new device, mathematical model is derived. Few papers have been dedicated only to mathematical modeling of different types of EH and provide crucial and useful information on this topic [47–49].

The mechanical model of the energy harvester set with a rotational pendulum can be represented by the equation of motion [50–52]. Into the equation of motion (4) for a pendulum the equation of motion correcting factor was introduced in order to reflect real situation from the experiment. To keep the rotational movement of the pendulum, the entire system with a vibration exciter was tilted, and for this angle φ is added to the equation of motion. We did not consider the equation for the magnetic

circuit. In our simplified model we took into account all the variables having an impact on pendulum's movement and they are contained in the following simplified equation of motion:

$$I\ddot{\varphi} + c\dot{\varphi} + m\omega^2 Ah\cos(\omega t)\sin\varphi + mgh\sin\varphi + F_{mag}sin(2\varphi + 1.5) = 0, \tag{4}$$

where: *I*—moment of inertia of pendulum, *c*—damping coefficient, *h*—distance from center of gravity to pivot axis, *m*—mass of pendulum and rotational additional masses, *g*—gravitation, *ω*—angular velocity of pendulum, *A*—amplitude of vertical excitation, $F_{mag}$—magnetic force amplitude acting on pendulum, *φ*—tilt angle of pendulum.

The phase shift of 1.5 rad present in the magnetic force term denotes the tilting of the pendulum (Figure 2). Here, we simplified the angular dependence of the magnetic force using sinusoidal dependence.

Some selected features of the Hamiltonian version of the considered model together with the separatrices which defines the regions of two basic solutions: oscillations and rotations are presented in Figure 9a–c. Note that the double well potential (Figure 9c) is generated by the interaction between the pendulum tip permanent magnet and additional permanent magnets distributed in the system as presented in Figure 2. In the studied system the magnetic forces are much stronger than gravity, which leads to a small splitting of the separatrices (see the differences of red and black lines in Figure 9b). The numerical results of the frequency sweeps are presented in Figure 10.

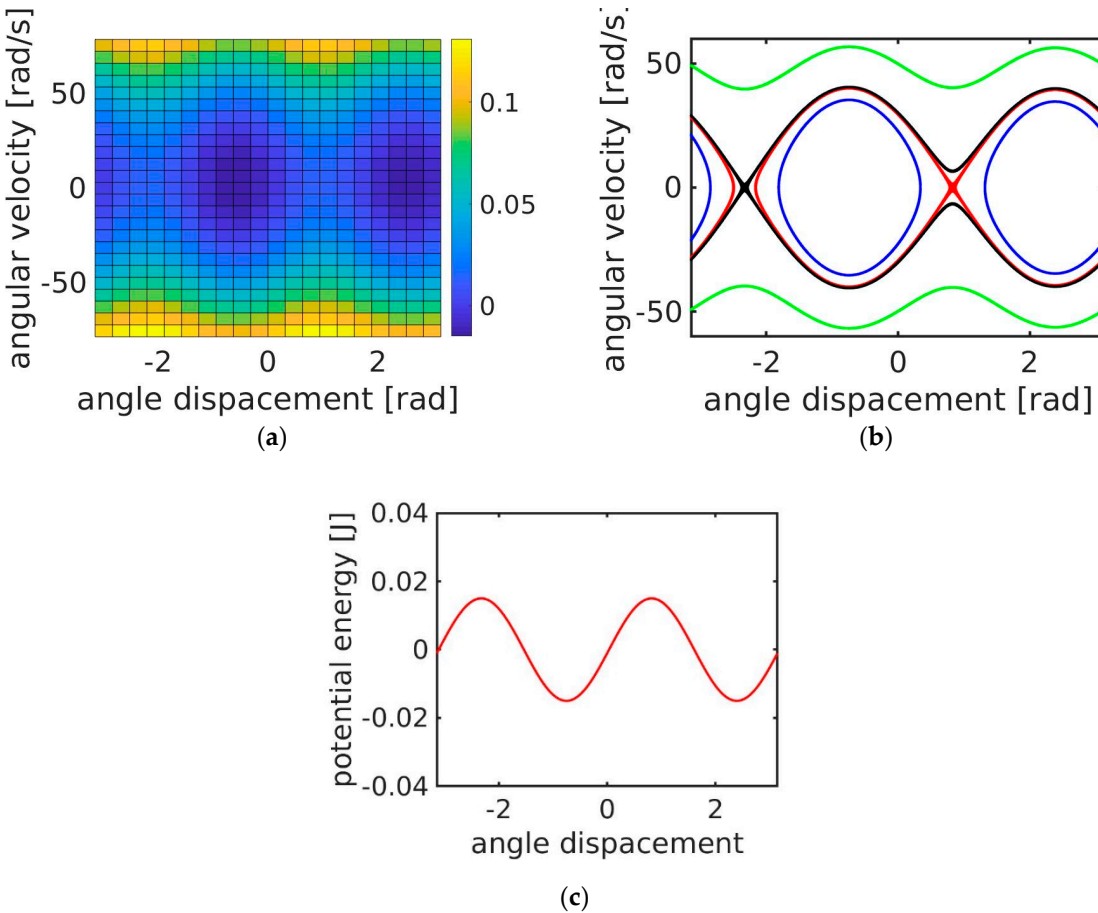

**Figure 9.** Selected model features for the mechanical resonator: (**a**) The total energy for the Hamiltonian system (without damping and denoted in colors [J] (see the color bar); (**b**) Possible solutions: oscillation (blue), rotation (green) and the separatrices—orbital lines crossing the saddle points (black and red); (**c**) Double well potential defined with in the pendulum angular variable [−π, π]. The system parameters are indicated in Table 4.

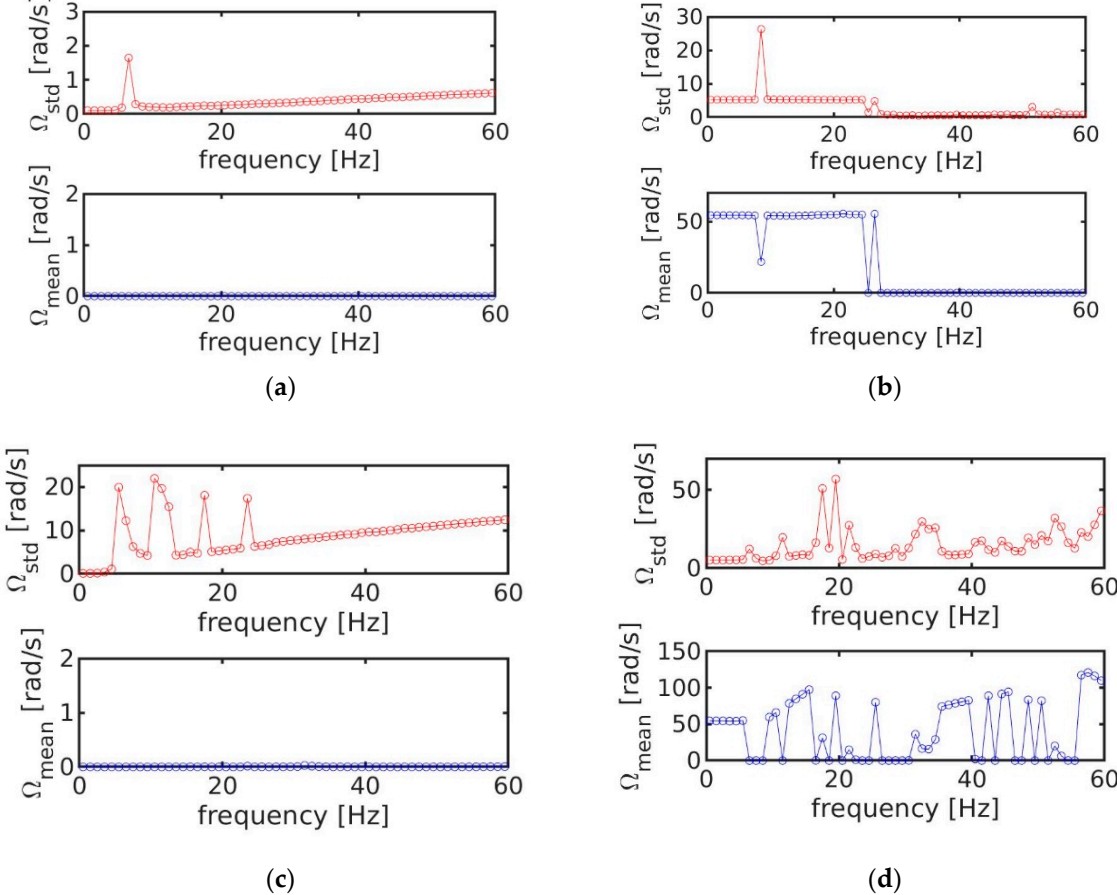

**Figure 10.** The frequency sweep results for a mechanical resonator. $\Omega_{std}$ and $\Omega_{mean}$ denote variability (standard deviation) and mean value (averaged in the time interval [100 s, 200 s]) of the rotational velocity versus excitation frequency. Here the amplitude of excitation was fixed at 2 mm (**a**,**b**) and 5 cm (**c**,**d**), while the initial conditions: angle $\varphi(t = 0) = 0$, velocity $\dot{\varphi}(t = 0) = 0$ (for **a**,**c**) and 80 (for **b**,**d**).

**Table 4.** Model system parameters used for numerical simulations (as written in Equation (4)).

| System Parameters | Values [Unit] |
|---|---|
| $h$ | 0.03325 [m] |
| $c$ | 0.000003 [N s/(kg m$^2$)] |
| $g$ | 9.81 [m/s$^2$] |
| $m$ | 0.00186 [kg] |
| $F_m$ | 0.03 [N] |
| $I$ | 0.0000377 [kg m$^2$] |
| $A$ | 0.002 or 0.05 [m] |

Figure 10 shows the frequency sweep results for pendulum response in its angular velocity. Our interest was focused on the angular velocity and the corresponding variability that are directly proportional to the electromotive forces in the electrical circuit. We used nodal initial conditions defined by the vertical position of the pendulum for initial angular displacement and two different initial conditions for angular velocity: zero and fairly large 80 rad/s. Two different excitation amplitudes were used, $A = 0.002$ and $A = 0.05$ m.

In Figure 10a we show the results of standard deviation and mean value for nodal initial velocity and small excitation. We noticed that the pendulum oscillates with small amplitude with visible resonance at the frequency of 6.5 Hz. This frequency corresponds to the natural frequency of the mechanical resonator. Simultaneously, the mean angular velocity is zero. In Figure 10b we present the

results of calculations repeated for large initial velocity and we observe another response in angular velocity. It remains finite up to frequency $f$ of about 24 Hz and collapses to the solution presented in Figure 10a. During the fairly large angular velocity response of 54 rad/s, the variability of this velocity was fairly small. Note that these results were obtained for the excitation amplitude which can be generated on the shaker. To complete the story, we enlarged the amplitude of excitation and obtained richer responses. In Figure 10b, we can observe a number of higher resonances which multiplies $f = 6.5$ Hz. This is a typical behavior for a non-linear system. Non-linearity based higher order resonances manifest with large enough excitation. On the other hand higher value of initial velocity drives the system to rotations. Interestingly, we observe a wide spectrum of rotational solutions separated by rotational instabilities leading to oscillatory vibrations. In addition, various rotational solutions appear. For small frequency $f < 6$ Hz we observe a similar dynamical response to the situation presented in Figure 10a. The output velocity is stable with changing frequency raging about 54 rad/s. This solution is related to overcoming the potential barrier (Figure 9c). Referring to Figure 9a,b it is easy to find that a velocity of 54 Hz is sufficient to overcome the potential barrier and obtain a rotational solution. This solution resembles the type of vibration coherence resonance [20], where excitation frequency is less important than the amplitude important to generate a large orbit motion with a minimum of excitation energy necessary to overcome the potential barrier. For higher frequency intervals such as [9 Hz, 15 Hz] and [35 Hz, 45 Hz], the response angular velocity is proportional to excitation frequency. Interestingly, for the first mentioned interval, proportionality is 1:1 while for the next it is 1:1/3. This is evidence that the solution has a sub-harmonic 3 property. In a summary of the results of Figure 10, one can conclude that the system has a great capability toward frequency broad band responses. In spite of the strong model simplifications, the simulations shows the large frequency region of rotational solutions. Such sub-harmonic solutions can be very useful for energy harvesting as proved by [17,53].

## 5. Conclusions

This paper presents an electromagnetic energy harvester with a rotational pendulum. The entire work presents a logical sequence from the system design, to the application of the finite element method to check magnetic interactions in the system, through testing and mathematical modelling. Performance of the FEM model showed magnetic interactions, which appear in the system and what are the expected magnetic forces by usage of co-energy. Results of calculated magnetic forces served to perform a simplified model of the rotational pendulum reflecting its kinematics in tests. Also, the application of the mentioned method let to us obtaining the surface of electromagnetic induction in domain of magnets' positions. Owing to this, three specific areas of interest were found depending on the position of additional magnets in coils. By the conducted tests, information about output voltage RMS and output power from the system was obtained. An experiment was performed with different values of operational frequencies and for different values of additional resistance, presenting in such a way a map of results. In the last stage, a simplified mechanical model of rotational pendulum was derived. To formulate the pendulum's equation of motion, an element from the experiment was added in the form of acting magnetic forces in the system and some phase shift was needed to start the pendulum's permanent rotational movement. Finally, simulations checked the application potential of the proposed device in the context of frequency broadness, which is important for variable ambient working conditions. However, one should remember that the large orbit (more energetic) is only one possible solution as a non-linear system is known for multiple solutions. To check the availability of the larger orbit solution, one needs to perform more studies including basins of attraction approach [54]. The system can be still optimized by changing its moment of inertia, the strength of magnets, improving the electrical circuit, and by carrying out the tests in wider range of frequencies. The next step in research on the system will be the creation of multiple potential wells, by increasing number of coils with magnets and studying multi-stable phenomena as in the oscillating energy harvesting systems [18,55].

**Author Contributions:** B.A.: methodology, validation, formal analysis, writing—original draft preparation, visualization, G.L.: methodology, investigation, software, formal analysis, supervision, P.W.: formal analysis, investigation, data curation, writing—review and editing, project administration, funding acquisition. All authors have read and agreed to the published version of the manuscript.

**Funding:** This publication was supported by the program of the Polish Ministry of Science and Higher Education under the project DIALOG 0019/DLG/2019/10 in the years 2019–2021. DIALOG

**Conflicts of Interest:** The authors declare no conflict of interest.

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
