# Peer review of "Modelling of Electromagnetic Energy Harvester with Rotational Pendulum Using Mechanical Vibrations to Scavenge Electrical Energy"

_applsci, doi:10.3390/app10020671_

Round 1
Reviewer 1 Report
The review of the State of the Art should be improved. Many examples of mechanical energy harvesters exploiting non linear mechanism, such as Snap Through Buckling, are deeply discussed in the literature.
Moreover, very detailed works exist which address the modeling of non linear mechanism for energy harvesting.
Authors should provide a deep review of the state of the art including linear and non linear energy harvesters.
The introduction should be restyled to better highlight main novelties introduced by this paper and advantages as respect to the SOTA.
Is the following statement the main motivation for this work?
"The motivation of taking up the topic was the need for create own system with application of new solutions, such as usage of non-magnetic material for housing and adjusted additional magnets inside coils."
Experimental set-up
More details should be given, especially concerning the accelerometer specification and the badwidth of the adoped shaking system.
Device modeling
Two sections have been dedicated to simulate and to model the mechanical behaviour of the device.
The matching between experimental results and the model proposed should be better introduced and discussed.
Authors should also provide comments related to the electrical modeling of the system.
Trends observed and reported in Fig.7 should be discussed and referred to the expected behaviour.
Reviewer 2 Report
This paper presented a study on the modelling of an electromagnetic energy harvester for harvesting electricity from mechanical vibrations. The authors used a 3D printed prototype for the experimental and used two software to model the outcome. This concept proposed is interesting. Pulication is recommended subject to minor revision according to the comments below and improvement in writing.
Comments to improve the manuscript:
As authors have used both numerical and experimental studies, the title can be amended to accordingly. The literature review is too short and insufficient. There are many proposed applications in similar fields (Piezoelectric and electrostatic) which can be addressed. Some relevant recent studies in piezoelectric energy harvesting can be added in line 47 page 2, such as:Izadgoshasb, I., Lim, Y. Y., Tang, L., Padilla, R. V., Tang, Z. S., & Sedighi, M. (2019). Improving efficiency of piezoelectric based energy harvesting from human motions using double pendulum system. Energy conversion and management, 184, 559-570.
Balguvhar, S., & Bhalla, S. (2019). Evaluation of power extraction circuits on piezo‐transducers operating under low‐frequency vibration‐induced strains in bridges. Strain, 55(3), e12303.
Izadgoshasb, I., Lim, Y. Y., Vasquez Padilla, R., Sedighi, M., & Novak, J. P. (2019). Performance enhancement of a multiresonant piezoelectric energy harvester for low frequency vibrations. Energies, 12(14), 2770.
Page 2 Line 59, please specify the dimension of the prototype, the material used, its weight and the source of vibration. Please mention the frequency range and the possible usage of this device in the real-life. Fig 4 caption needs to be adjusted (large font is used). According to Table 1, please explain why the frequency range is chosen between 10 and 15 Hz? Is the resonant frequency of system calculated? In the conclusion, please compare the results with former studies. What improvement is achieved? Please improve this section. Writing needs to be significantly improved.
Reviewer 3 Report
Authors should modified the title of article. The main focus should be in rotational pendulum.
Abstract has a not a good aspect
Introduction chapter. The novelty of the work is not clarified and literature review does not support this goal
Authors should mention why they chose this system (12 Inc. etc.) and what are the main differences from the typical ones (Energy Harvester systems). Also authors should deeply analyze the advantages and disadvantages of presented system.
Methodology chapter. Methodology is very weak. It’s difficult understand how experiment is related with simulation actions? Authors should deeply present the main characteristics of equipment or show the links where we could recognize it. The fig.2 – why author show the accelerometer if they don’t show any result of it measures? What initial conditions describes the FEM modelling? Fig.4 – where is the x axis? In my opinion authors should show it in the previous figures.
Results. Figures are very helpful but the comparison with existing literature? In addition, where is the critical analysis of results? Could authors deeply analyze the Figure 8, its difficult understand what the main idea authors want to present.
Conclusions. Authors should write conclusion focused in the novelty of getting results. In my opinion it’s not necessary to write one more abstract in the conclusion chapter. Authors must modified it accord this remark.
Round 2
Reviewer 3 Report
I have no remarks.
Author Response
We would like to thank you for the acceptance of introduced revisions of the article. We see the potential in our energy harvester and we are going to perform next experiments with it. We hope that the written article will be good quick reference both for experienced and inexperienced scientists dealing with energy harvesting.